

# A year of transient tracers chlorofluorocarbon 12 and sulfur hexafluoride, noble gases helium and neon, and tritium in the Arctic Ocean from the MOSAiC expedition (2019-2020)

Céline Heuzé[1], Oliver Huhn[2], Maren Walter[2,3], Natalia Sukhikh[2,4], Salar Karam[1], Wiebke Körtke[2], Myriel Vredenborg[5], Klaus Bulsiewicz[2], Jürgen Sültenfuß[2], Ying-Chih Fang[6], Christian Mertens[2], Benjamin Rabe[5], Sandra Tippenhauer[5], Jacob Allerholt[5], Hailun He[7], David Kuhlmey[5], Ivan Kuznetsov[5], and Maria Mallet[5]

[1]Department of Earth Sciences, University of Gothenburg, Gothenburg, Sweden
[2]Institute of Environmental Physics, University of Bremen, Bremen, Germany
[3]MARUM, Center for Marine Environmental Sciences, University of Bremen, Bremen, Germany
[4]Lomonosov Moscow State University Marine Research Center, Moscow, Russia
[5]Alfred-Wegener-Institut Helmholtz-Zentrum für Polar- und Meeresforschung, Bremerhaven, Germany
[6]Department of Oceanography, College of Marine Sciences, National Sun Yat-sen University, Kaohsiung, Taiwan
[7]State Key Laboratory of Satellite Ocean Environment Dynamics, Second Institute of Oceanography, Ministry of Natural Resources, Hangzhou, China

**Correspondence:** Céline Heuzé (celine.heuze@gu.se)

**Abstract.** Trace gases have demonstrated their strength for oceanographic studies, with applications ranging from the tracking of glacial meltwater plumes to estimates of the abyssal overturning duration. Yet measurements of such passive tracers in the ice-covered Arctic Ocean are sparse. We here present a unique data set of trace gases collected during the Multidisciplinary drifting Observatory for the Study of Arctic Climate (MOSAiC) expedition, during which the R/V Polarstern drifted along with the Arctic sea ice from the Laptev Sea to Fram Strait, from October 2019 to September 2020. During the expedition, trace gases from anthropogenic origin chlorofluorocarbon 12 (CFC-12), sulfur hexafluoride ($SF_6$), and tritium, along with noble gases helium and neon and their isotopes were collected at a weekly or higher temporal resolution throughout the entire water column and occasionally in the snow, from the ship and from the ice. We describe the sampling procedures along with their challenges, the analysis methods, the data set, and present case studies in Fram Strait and the Central Arctic Ocean to illustrate possible usage for the data along with their robustness. Combined with simultaneous hydrographic measurements, this trace gases data set can be used for process studies and water mass tracing throughout the Arctic in subsequent analyses.

## 1 Introduction

The full-depth Arctic Ocean is changing rapidly in response to ongoing climate change (Meredith et al., 2019). In the upper ocean, the sea ice cover is thinning and reducing, overall (e.g. Kwok, 2018), while freshwater input from glaciers, rivers and the atmosphere (Solomon et al., 2021) and heat input from the global ocean (Polyakov et al., 2020) are changing. These factors have resulted notably in contrasting changes in the stratification in the Arctic basins (Polyakov et al., 2018) and in



an intensification of the Beaufort Gyre (Timmermans and Toole, 2023). As the exact processes responsible for these changes remain unclear, so does the future of the upper Arctic Ocean (Muilwijk et al., 2023). In the deeper layers of the Arctic Ocean, we do not even know whether there is a change, as hydrographic observations deeper than 1000 m are too sparse in space and
20   time for proper dynamics studies (Heuzé et al., 2022). There is an urgent need to establish a baseline for the under-observed full depth Arctic Ocean circulation, including its spatial and temporal variability, and observe changes in near-real time. We argue that passive tracers, as presented in this paper, are the ideal tool to not only increase data coverage in the Arctic, but also to study the processes that impact the Arctic Ocean.

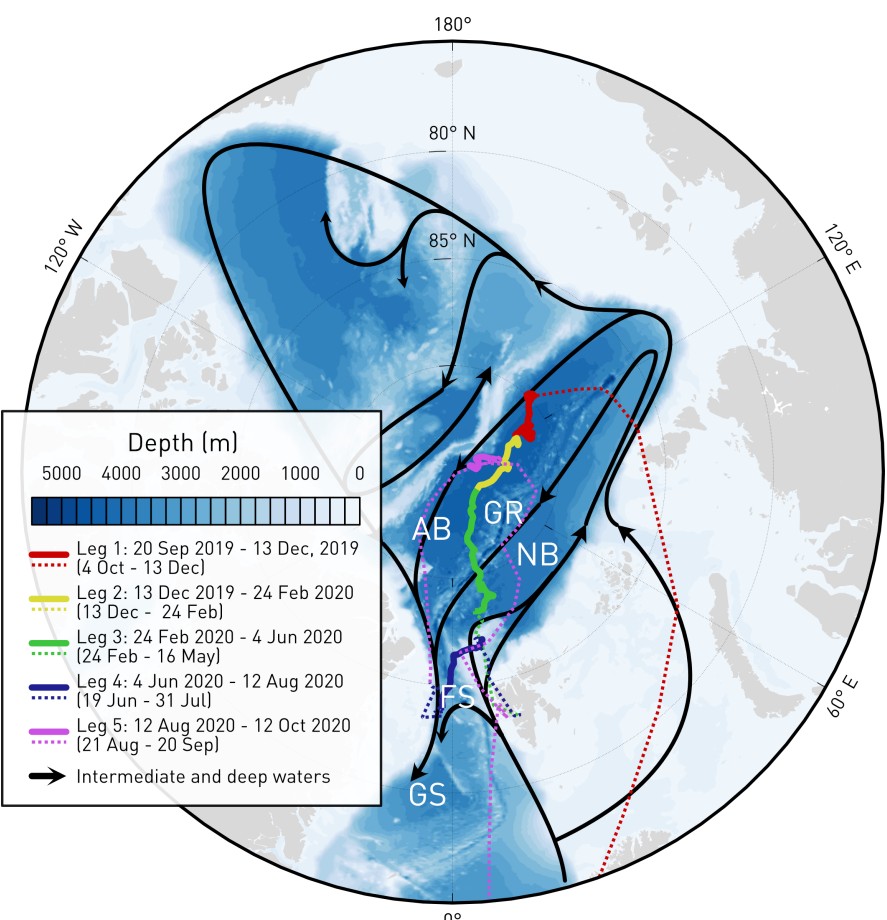

**Figure 1.** Track of the five legs of the MOSAiC expedition. Solid lines show when Polarstern was drifting with the sea ice, and dotted lines show the transit. Dates for each leg are given, with dates excluding transit in parentheses. Black arrows show the main circulation features of the Arctic Ocean intermediate (Atlantic) and deep waters. The place names discussed in the text are Greenland Sea (GS), Fram Strait (FS), Nansen Basin (NB), Gakkel Ridge (GR), and Amundsen Basin (AB). The bathymetry (blue-white colours) is from the International Bathymetric Chart of the Arctic Ocean (IBCAO; Jakobsson et al., 2020), and the land mask is from A Global Self-consistent, Hierarchical, High-resolution Geography Database (GSHHG; Wessel and Smith, 1996).



To study the full-depth ocean circulation, we need prolonged measurements over a large area at relatively high spatial and
temporal resolutions. In the rest of the world ocean, thousands of autonomous profilers have been monitoring the upper 2000 m
since the 1990s (Johnson et al., 2022). Although ice-avoidant and/or ice-tethered profilers have been deployed in the Arctic
(Toole et al., 2011), their uninterrupted operation remains a challenge in the ice-covered ocean. Besides, they are limited to
the upper 1000 m; to the best of our knowledge, no full-depth autonomous profiler has been deployed in the Arctic Ocean
yet. Another option is to use trace gases, which has been done since the beginning of modern-day Arctic research (e.g. Top
et al., 1983; Schlosser et al., 1990). The trace gases that we focus on here are all passive tracers, that is, they are not affected
by chemical or biological activity. Consequently, by comparing their concentration throughout the water column or between
profiles, one can infer the processes that have affected the water, the water circulation, and even the age of the water. These
tracers have different sources and, therefore, different applications, schematically represented in Fig. 2.

The inert, stable noble gases helium (He) and neon (Ne) are abundant in the atmosphere but have a low solubility. Con-
sequently, supersaturation at the surface indicates that processes that inject air bubbles, for example wave breaking or wind,
have recently taken place (e.g. Hahm et al., 2004). Below the surface, excess helium and neon allow for the detection and
even quantification of glacial/basal melt water (e.g. Schlosser, 1986; Beaird et al., 2015; Huhn et al., 2021): Atmospheric air
with a constant composition of these noble gases is trapped in the ice matrix during formation of the meteoric ice. Due to the
enhanced hydrostatic pressure at the base of the shelf ice, these gases are completely dissolved in the water, when the ice is
melting from below. This leads to an excess of He of 1280% and Ne of 890 % in pure melt water (Loose and Jenkins, 2014).
Besides, glacial meltwater can be enriched by crustal $^4$He, leading to anomalously high He/Ne ratios in the relative vicinity of
Greenland fjords (Beaird et al., 2015; Huhn et al., 2021). In the Central Arctic, the He/Ne ratio at the surface is a proxy for sea
ice processes as the noble gases fractionate during sea ice formation: The lighter He is incorporated in the ice matrix, whereas
Ne is rejected along with the brine. Anomalously low He/Ne can therefore indicate recent sea ice formation (e.g. Top et al.,
1983; Hahm et al., 2004).

The helium isotope $^3$He has its main source in the Earth's interior, the mantle and crust. This primordial helium gets injected
into the deep ocean via the hydrothermal circulation of seawater through the crust, which leads to an excess in $^3$He compared
to the atmospheric equilibrium value of the isotopic ratio found in the upper ocean. Thus, the isotopic ratio (specifically $\delta^3$He,
the excess $^3$He compared to the atmospheric ratio) can be used as a tracer for hydrothermal venting (German et al., 2022) and
the vertical exchange between the interior ocean and the upper layers (Rhein et al., 2010). The other source of excess $^3$He is
tritiugenic, i.e. it is produced by the decay of tritium. Tritium ($^3$H) is the radioactive isotope of hydrogen, and enters the ocean
as a result of nuclear testing in the 1960s via meteoric water from local precipitation and continental runoff, making it an ideal
tracer for studying the penetration of surface waters into the deep. Its half life is 12.32 years, and simultaneous measurements
of tritium and its decay product $^3$He can also be used as an age tracer.

Chlorofluorocarbon 12 (CFC-12) and sulfur hexafluoride (SF$_6$) are anthropogenic trace gases with well known atmospheric
concentrations (Bullister, 2015; Dutton et al., 2022a, b). The gases are well mixed in the atmosphere with only a small dif-
ference between the northern and southern hemisphere. The atmospheric CFC concentrations increased exponentially in the
1970s, linearly afterwards, and the growth rate started to decrease after the Montréal Protocol of 1987. CFC-12 reached its



peak in atmospheric concentration in 2002/03. For $SF_6$ the increase still continues and thus it can be used for younger, more
60  recently ventilated waters. For the ocean, the atmosphere is the only source of the trace gases CFC-12 and $SF_6$, since there
are no significant natural sources. The gases enter the surface waters of the ocean through air-sea gas exchange and can reach
equilibrium concentration with the atmosphere. However, especially in the Arctic Ocean and for $SF_6$, a 100% saturation is
normally not reached due to a too slow adaption to changing conditions (e.g. Smith et al., 2022; Tanhua et al., 2009).

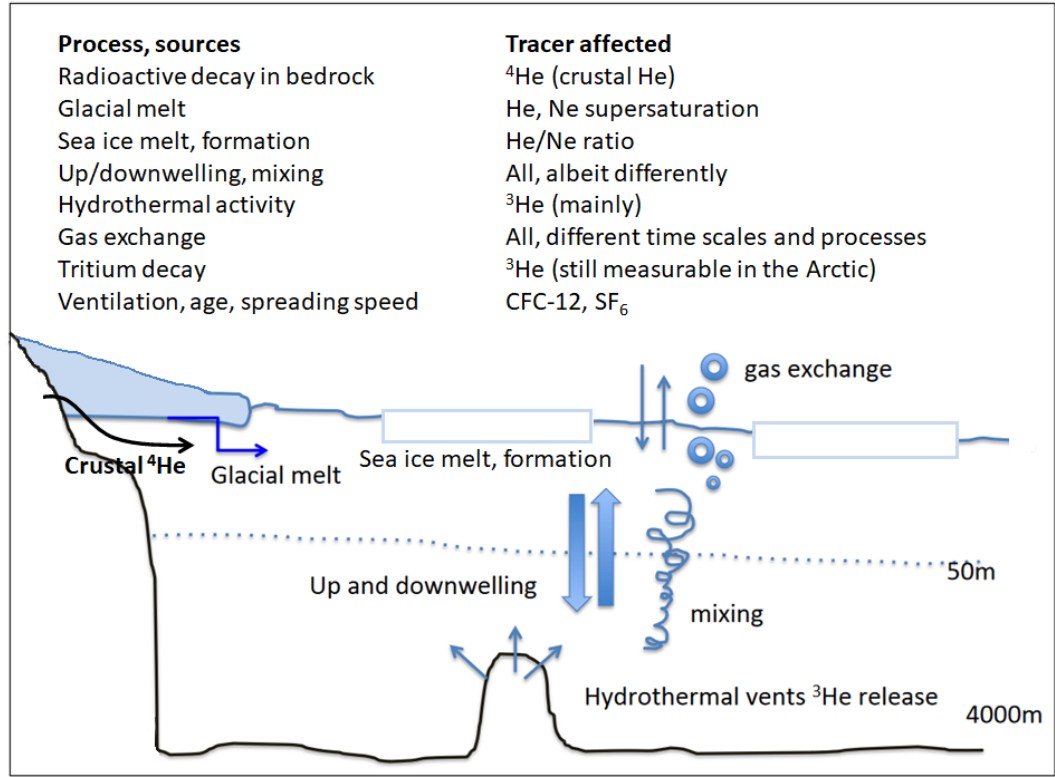

**Figure 2.** Schematic of the ocean-ice-atmosphere interactions affecting the tracers sampled during MOSAiC.

We report on trace gases measurements of CFC-12, $SF_6$, helium, neon, and tritium made as part of the physical oceanography
65  programme of the Multidisciplinary drifting Observatory for the Study of Arctic Climate (MOSAiC) expedition. From October
2019 to September 2020, the German icebreaker R/V Polarstern (Knust, 2017) drifted with the sea ice pack within the Eurasian
Arctic (Fig. 1) and served as a scientific platform, allowing the collection of water samples throughout the entire water column
even in the middle of winter. The expedition and overall physical oceanography programme are described in detail in Rabe
et al. (2022). We here describe the sampling strategy in the field in section 2, including some of the challenges encountered
by our team. Laboratory analyses are described in section 3. The data set is briefly described in section 4. In section 5, we
demonstrate the validity of our data and show possible applications using samples from two distinct regions: Fram Strait and





the Central Arctic. We conclude in section 5 with a brief discussion of the wider scope of the data set, and the lessons learnt from MOSAiC.

## 2    Sampling strategy during MOSAiC

There were two main aims for the tracer sampling during MOSAiC:

1) Upper ocean (down to the subsurface warm and salty Atlantic Water layer, hereafter referred to as "Atlantic Water"): The goal is a better understanding of the mixed layer processes in the horizontal and vertical, with a focus on the role of the sea ice and the proximity of the ice edge, and how it affects the exchange of heat between atmosphere, mixed layer, and the heat sources in the interior (i.e. the Atlantic Water). The combination of the anthropogenic tracers (CFC-12, $SF_6$) with the noble gas
isotopes and tritium is used to study the integral effect of events like leads, eddies, and storms on the mixed layer properties and the vertical exchange between the mixed layer and the Atlantic Water.

2) Deep ocean (from Atlantic Water to the sea floor): The goal is to determine the deep oceanic circulation in the Arctic, notably which route(s) the deep waters take, the ventilation processes, and the age of the waters. Only CFC-12 and $SF_6$ were sampled for this; future efforts to track the ventilation should also sample tritium, while measurements above Gakkel Ridge of
helium would be ideal.

This two-fold purpose resulted in the following general sampling strategy: During the weekly hydrographic casts (Conductivity-Temperature-Depth or CTD) from the ship (Rabe et al., 2022), tracer samples were collected from a water bottle rosette over 12 depth levels covering the entire water column (circles on Fig. 3). Additional sampling over the upper 500 m took place from CTD casts in the ice camp (Ocean City or "OC"), mainly during spring (stars on Fig. 3). Trace gas samples were the first to
be collected at the rosette to avoid potential degassing. Prior to sampling, the metal tubing and intake adapter were cleaned with isopropanol to remove any fat, and the person sampling made sure to not directly touch these parts. We now describe the procedure for each specific sampling, and provide a list of selected challenges encountered.

### 2.1    Sampling of helium and neon; of tritium; and of CFC-12 and $SF_6$

In total we took 290 water samples for stable noble gas isotopes ($^3$He, $^4$He, $^{20}$Ne, $^{22}$Ne) during Legs 1-5 (purple and yellow
on Figs 3 and 4). The water samples were stored from the CTD/water bottle rosettes (ship and Ocean City) without contact to atmospheric air into  40 ml gas tight copper tubes, which are clamped off at both sides. They were collected straight after the CFC-12 / $SF_6$ samples if at the same rosette bottle, and collected first if no transient tracer sample was needed at that bottle. The person sampling took great care to rid the plastic tubing for sampling of any bubble by letting the water flow for as long as necessary, and by regularly hitting the copper tube with a wrench.
For tritium measurements we took 143 sea-water samples during Legs 1-5 (yellow on Figs 3 and 4). The samples were stored in 500 ml plastic water bottles from the CTD/water bottle rosettes (ship and Ocean City). Additionally, we opportunistically took 9 samples from snow into 2x500 ml plastic bottles during Leg 3 (diamonds in Fig. 3).



**Figure 3.** Depth and date of all the samples included in this data set. Note the discontinued y-axis. Red shows where only CFC-12/SF₆-data were collected, purple where noble gases were additionally sampled, and yellow dots where tritium was sampled in addition to all other tracers. Circles indicate that the sample was collected from the ship; stars, from Ocean City. Diamonds above the figure indicate the dates in March to May 2020 (Leg 3) of the tritium from snow samples.

For the transient tracers CFC-12 and SF₆, we took 410 samples during Legs 1-5, all the way to the sea floor (red, purple and yellow on Figs 3 and 4). The CFC-12 and SF₆ water samples from the CTD-bottle systems were stored in 220 ml glass
ampoules, avoiding contact to the atmosphere during the tapping by a dedicated tubing and rinsing procedure. After sampling, the ampoules were flame sealed after a headspace of pure nitrogen had been applied. Flame-sealing started immediately after the sampling, but due to the large number of samples and the fact that only one sample could be sealed at a time, up to 6 hours passed between sampling and the sealing of the last ampoule.

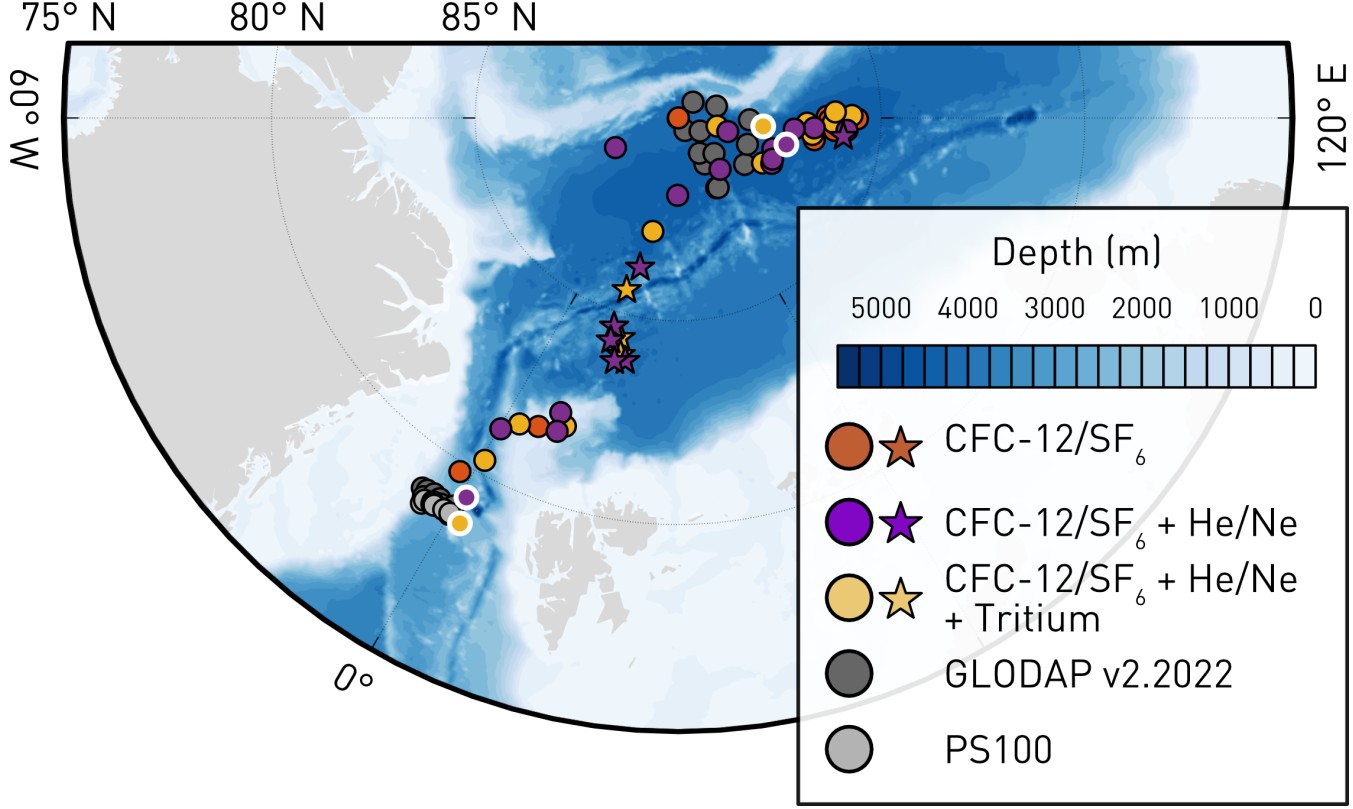

**Figure 4.** Location of tracer data collected during the MOSAiC expedition. Red colours show where only CFC-12/SF$_6$-data were collected, purple colours where He/Ne were additionally sampled, and yellow colours where tritium was sampled in addition to all other tracers. The circles denote samples taken from the ship, and the stars from Ocean City. The location of the four example casts analysed in section 5 are denoted by thick, white outlines, and the reference values to which they are compared with grey dots (see text). Bathymetry (blue-white colours) is from IBCAO (Jakobsson et al., 2020); land mask from GSHHG (Wessel and Smith, 1996).

## 2.2 Challenges encountered in the field

During the MOSAiC expedition, the team met challenges during each leg that affected the sampling strategy. The difficulties can be divided into three categories: organisational, instrumental, and strategical. The first includes planning and training issues. The limited overlap time between legs reduced the experience exchange between groups. Additionally, most team members were new to tracer sampling and had only training in the lab on shore, therefore they had to train at the beginning of each leg and adapt on-site to sampling under (occasionally) extreme conditions. The small teams also had many other tasks, which led

to sample labelling errors that had to be identified and corrected post cruise in the lab. At a higher organisational level, the initial plan to bring samples back after each leg was abandoned due to logistical problems and harsh conditions, thus causing delays in lab analysis. Finally, the Covid-19 pandemic dramatically altered the expedition and sampling plans. The team of Leg 3, smallest of all legs as they should have spent the shortest time there, had to stay onboard for nearly 4 months, as an exchange

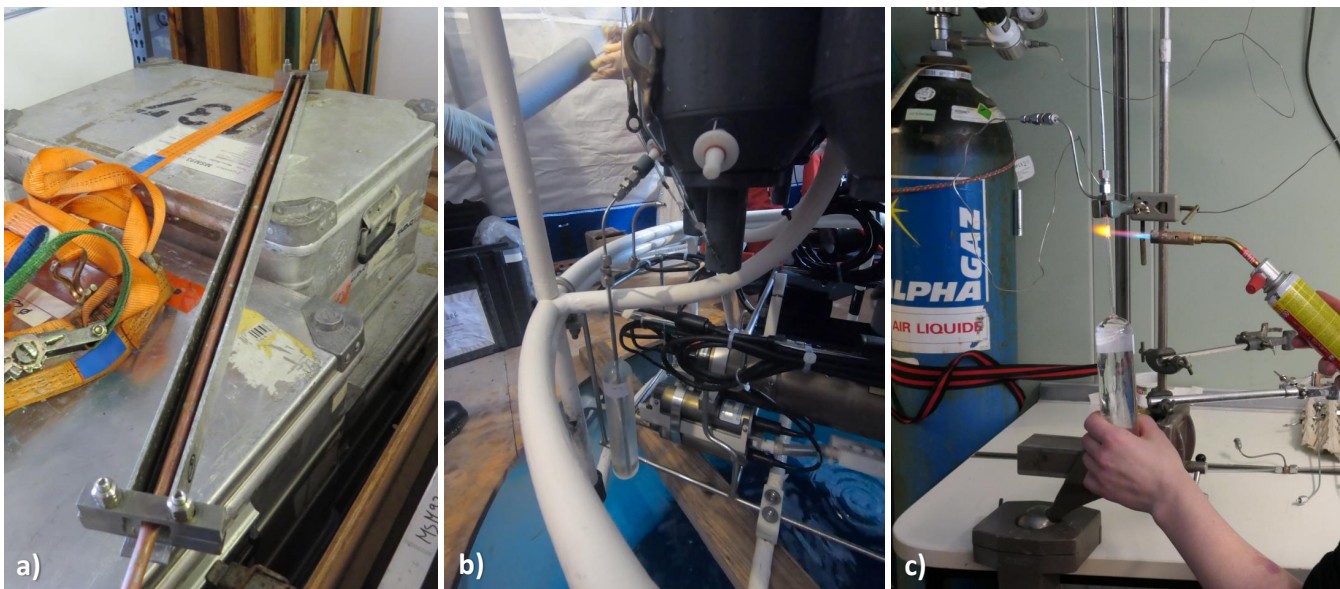

**Figure 5.** Pictures illustrating sampling challenges: a) 90 cm-long rigid copper tube for noble gas sampling (Credit: Wiebke Körtke); b) Sampling of CFC-12 and $SF_6$ during Leg 3 from "Ocean City", tent on the sea ice, with the deep ocean directly below our ampoules (Credit: Natalia Sukhikh); c) Flame sealing of the CFC-12 sample onboard the R/V Polarstern (Credit: Janin Schaffer).

of crew and scientists was hampered. They were, understandably, under a lot of stress, and with reduced communication with
offices and labs on land that were locked down.

The second group of challenges is related to instrumental problems. The set-up for the equipment and the lab space for the fixing and storing of the samples had to be changed several times, which took time and effort. Besides, we suspect that changes in storage conditions or movement of the flame sealing system are the cause for the blockage of the pipe from the nitrogen bottle at the beginning of Leg 3. The pipe connecting the nitrogen bottle and sampling ampoules was blocked by an unknown
obstacle. The blocked part of the pipe was eventually detected and removed, but as it was impossible to resolve that issue with the team onboard, the communication with the office on land took several days and led to more delays.

Finally, the sampling strategy had to be adapted during Leg 3 due to the loss of the ship-CTD hydrohole by compressed ice (Rabe et al., 2022). Combined with the uncertainties from the development of Covid-19, it was decided that the ship would not be moved to open a new hole. It was therefore impossible to sample from the ship and perform deep water sampling
from March to May, when the ship eventually moved to exchange personnel. Consequently, key regions of interest such as the Amundsen-Nansen basin transition and the Gakkel Ridge were not covered with deep samples (stars on Figs. 3 and 4). Sampling continued from Ocean City, the tent on the ice equipped with a small CTD/water sampling system. However, the OC was not well suited for CFC-12/$SF_6$ and noble gas sampling. There was not enough space for these operations (the copper tubes are 90 cm long and rigid) and the volume of bottles was significantly smaller than for the ship-CTD bottles (12 instead of
24 bottles, with 5 L instead of 10 L volume, to be shared with the other teams). Transportation of the water samples in fragile



glass ampoules from the ice to the ship's lab in freezing temperatures proved to be the most challenging part. Solutions for a closing hydro-hole and safely transporting samples on the ice had been discussed by the whole MOSAiC consortium before the expedition. Unfortunately, funding was not available nor prioritised to implement the solutions.

## 3 Analysis, calibration, and validation of the samples in the lab

Noble gas samples and flame-sealed transient tracer samples were stored onboard until the end of the expedition. They were then analysed at the Institute of Environmental Physics (IUP) of the University of Bremen, Germany, following standard procedures (Bulsiewicz et al., 1998; Sültenfuß et al., 2009), as described in the following subsections.

### 3.1 Helium and neon samples

In the IUP Bremen noble gas lab the samples were pre-processed with a UHV (ultra high vacuum) gas extraction system.
Sample gases are transferred via water vapour into a glass ampoule kept at liquid nitrogen temperature. For analysis of the noble gas isotopes the glass ampoules are connected to a fully automated UHV mass spectrometric system equipped with a two stage cryogenic system and a quadrupole and a sector-field mass spectrometer. Regularly, the system is calibrated with atmospheric air standards (reproducibility $< 0.2\%$). Measurement of line blanks and linearity are done as well. The performance of the Bremen facility is described in Sültenfuß et al. (2009).

Noble gas concentrations are reported in nmol/kg (for total He = $^4$He + $^3$He, and total Ne = $^{20}$Ne + $^{22}$Ne) or percent (for $^3$He) such as:

$$\delta^3 He = 100 \times \frac{[^3He/^4He]_{water} - [^3He/^4He]_{air}}{[^3He/^4He]_{air}}. \tag{1}$$

However, for presentation in this paper, we use for total He and total Ne the gas excess in percent:

$$\Delta He = 100 \times \frac{He_{water} - He_{equilibrium}}{He_{equilibrium}} \tag{2}$$

using the equilibrium functions $He_{equilibrium}$=f(T,S) and $Ne_{equilibrium}$=f(T,S) from Weiss (1971), where T and S are the potential temperature and practical salinity as recorded at the Niskin bottle. The advantage of this common unit for noble gases is that it "removes" the equilibrium concentration caused by atmospheric gas exchange for the given T and S and shows only the gas-excess caused by, e.g., bubble injection, basal glacial meltwater, hydrothermal addition, or other processes inside the ocean. Ultimately, 208 samples were analysed successfully, including 25 pairs of replicate samples that were each averaged for the
final data set. The precision is 0.4% for He, 0.7% for Ne, and 0.8% for $\delta^3$He (based on the 25 pairs of replicate measurements). 22 samples were flagged doubtful; these error flags are based on comparison with other properties and identification as outliers.

### 3.2 Tritium samples

In the IUP Bremen noble gas lab the water samples were pre-processed with a gas extraction system for complete degassing and were then stored for several months. During that time, part of the tritium ($^3$H) decayed by beta-decay to helium 3 ($^3$He).

Earth System
Science
Data

The new produced $^3$He was than analysed by the same mass spectrometer system as described above. Tritium concentrations reported here are scaled to 1 January 2020 and referred to as TU2020. Concentrations are given in TU (tritium unit), where 1 TU is the ratio of 1 tritium atom to $10^{18}$ hydrogen atoms. Typical errors for this data set are 0.04 TU or 3%, whichever is largest.

### 3.3  CFC-12 and SF$_6$ samples

The determination of CFC-12 and SF$_6$ concentrations in the IUP Bremen gas chromatography lab is accomplished by purge and trap (cryogenic trapping at -65°C) sample pre-treatment of a precise water volume of 140 ml followed by gas chromatographic separation on a capillary column and electron capture detection (ECD). After thermal desorption the released gases are separated on a pre-column of type Aluminia Bond/CFC, 0.54 mm ID x 3m, and a main column of type Aluminia BOND/CFC, 0.54 mm ID x 30 m. SF$_6$ and CFC-12 are then detected on a micro-ECD.

The analytical system is calibrated frequently by analyzing different volumes of known standard gas concentrations. The loss of CFC-12 and SF$_6$ into the headspace is considered by equilibration between liquid and gas phase under controlled conditions before the sealed ampoules are opened and the volume of the headspace precisely measured. A more detailed description of the measurement system is given by Bulsiewicz et al. (1998).

CFC-12 concentrations are reported in pmol/kg and SF$_6$ in fmol/kg, both reported on SIO98 scale (Prinn et al., 2000). We
use these units to show the data in this paper. 271 samples were analysed successfully, including 43 pairs of replicate samples that were each averaged for the final data set. The precision of the measurement, based on the comparison of the replicate samples, is 1% or 0.003 pmol/kg for CFC-12 (whichever is largest) and 2% or 0.02 fmol/kg for SF$_6$ (whichever is largest). The accuracy for CFC-12 is 2% or 0.005 pmol/kg (whichever is largest) and for SF$_6$ is 3% or 0.03 fmol/kg (whichever is largest), including errors of calibration, linearity, standard-gas, gas volumes for calibration, water volume, gas loss into the head-space,
and calibration scale. 7 samples for CFC-12 were flagged doubtful and 2 were flagged bad. 8 samples for SF$_6$ were flagged doubtful and 6 were flagged bad. These error flags are based on either suspicious processing during the measurement (e.g., failure during cryogenic trapping or others) or by comparison with other properties and identification as outliers.

## 4  Structure of the data set

Data and metadata of all samples where at least one of the gases was successfully analysed are provided as a single ASCII
(.dat) file. The data set has been submitted to PANGAEA (https://www.pangaea.de/). As per the MOSAiC data policy, it will be freely available there, without embargo, as soon as the technical checks have been completed by the PANGAEA team.

The data set metadata are identical to those of the MOSAiC CTD data sets, also on PANGAEA, to facilitate cross-analysis. These are (see Table. 1):

   – The station, or MOSAiC week, and cast numbers;

– The MOSAiC Leg number;



**Table 1.** Summary of the data included in the data set. Station, cast, leg, date, latitude and longitude, bottle number and bottle pressure are the same as in the MOSAiC CTD data sets. The World Ocean Circulation Experiment (WOCE) flags are: 2 = good; 3 = doubtful; 4 = bad; 6 = mean of replicates; 9 = no measurement.

| Parameter | Unit | Short description |
|---|---|---|
| Station | - | Station = number of weeks since MOSAiC drift started |
| Cast | - | Cast number for that station |
| Leg | - | Leg number; see Fig. 1 |
| ID | - | 1 if sample taken from the ship; 2 from OC; 3 in snow |
| Year | date (UTC) | CTD cast start year as recorded by the DShip system |
| Month | date (UTC) | CTD cast start month as recorded by the DShip system |
| Day | date (UTC) | CTD cast start day as recorded by the DShip system |
| Latitude | degree N | CTD cast start latitude as recorded by the DShip system |
| Longitude | degree E | CTD cast start latitude as recorded by the DShip system |
| Bottle number | - | Niskin bottle from which the sample was drawn |
| Bottle pressure | dbar | Pressure as recorded by the Niskin bottle |
| CFC-12 | pmol/kg | CFC-12 concentration |
| CFC-12-Flag | - | WOCE flag for CFC-12 (see caption) |
| SF6 | fmol/kg | $SF_6$ concentration |
| SF6-Flag | - | WOCE flag for $SF_6$ (see caption) |
| Helium | nmol/kg | Total helium concentration (primarily $^4$He) |
| Helium-Flag | - | WOCE flag for helium (see caption) |
| d3He | % | $\delta^3 He = 100 \times \frac{[^3He/^4He]_{water} - [^3He/^4He]_{air}}{[^3He/^4He]_{air}}$ (see section 3.1) |
| d3He-Flag | - | WOCE flag for $\delta^3 He$ (see caption) |
| Neon | nmol/kg | Total neon concentration ($^{20}$Ne and $^{22}$Ne) |
| Neon-Flag | - | WOCE flag for Neon (see caption) |
| Tritium | TU2020 | Tritium concentration scaled to 1 January 2020 (see section 3.2) |
| Tritium-Flag | - | WOCE flag for Tritium (see caption) |

- The start date of the CTD cast;

- The start latitude and longitude of the CTD cast;

- The Niskin bottle number, and its recorded pressure.

We also provide an ID variable to indicate whether the sample was collected from the ship or OC CTD (two different CTD
data sets), or in the snow. The snow data have "snow" as station name; cast, leg and bottle number set to the missing value -9;
and the bottle pressure set to 0. The data set contains first all ship data, then all OC data, and finally the snow data.



For all variables we use the quality flags of the World Ocean Circulation Experiment (WOCE), where 2 indicates a good value; 3, doubtful; 4, bad; 6, that the value is the mean of several replicates; and 9 that there is no measurement.

## 5    Example usage of the data

In this section, we verify that our measured values are sensible and demonstrate possible applications of these tracers for scientific studies. We show four full-depth profiles, two in Fram Strait collected late July / early August 2020, and two in the Central Arctic Ocean collected a month later. For each region we compare the profiles to each other and to historical values, and finally compare the two regions.

Profiles for the historical comparison were selected from the Global Ocean Data Analysis Project (GLODAPv2, Lauvset
et al., 2022, dark grey on Fig. 4). In Fram Strait, the main criterion was to remain in the deep parts of the Greenland Sea, i.e. east of the 500 m isobath and west of the Prime meridian. The GLODAP profiles are at most 50 km from the centre coordinate of the two studied MOSAiC profiles in subsection 5.1. As GLODAPv2 has no noble gas data in Fram Strait, we compare our values to those collected during PS100 (light grey on Fig. 4), published in Huhn et al. (2021). In the Central Arctic where historical values are rarer, we selected all CFC-12 and SF$_6$ profiles in GLODAP that are within 200 km of those studied in
subsection 5.2, in the deep Amundsen basin (depth > 3500 m). This yielded 12 profiles to compare to. To our knowledge, no public domain data set for noble gases contains data in the Central Arctic, and therefore limit our comparison to values from the literature. Similarly, although we acknowledge the existence of tritium data in the Arctic in the Jenkins et al. (2019) data set, we cannot use them for direct quantitative comparison as they have been decay-corrected to 1997, i.e. approximately two tritium half-lives ago.

To facilitate the discussion, we also provide the corresponding full depth Conservative Temperature profiles as well as the Conservative Temperature - Absolute Salinity (TS) diagrams (Fig. 6). More information about these variables can be found in the MOSAiC OCEAN overview (Rabe et al., 2022), which also details how to derive the mixed layer depth and Atlantic Water properties. All profiles have a shallow mixed layer not exceeding 10 m, which is to be expected for summer profiles. The Atlantic Water core (temperature maximum around 200 dbar on Fig. 6a and c, or peak to the right of b and d) is deeper and
colder for the Central Arctic profiles than for the Fram Strait ones. We therefore expect tracers to show that the Atlantic Water is older in the Central Arctic Ocean than in Fram Strait. The Fram Strait casts are supposedly in the Arctic outflow, i.e. should be the oldest, but instead are most likely recirculating young water. We will discuss this further in subsection 5.3. The casts of Fram Strait have many intrusions in their upper 200 m, which we discuss in the next subsection. The strong difference in surface salinity between the two casts of the Central Arctic (bottom left of Fig. 6d) is discussed in subsection 5.2. Although this
does not affect our results, the reader should bear in mind that the hydrography is that of the downcast, when the water column is least perturbed, but the samples were collected during the upcast. Especially in layers with active mixing or intrusions, the two do not match perfectly.

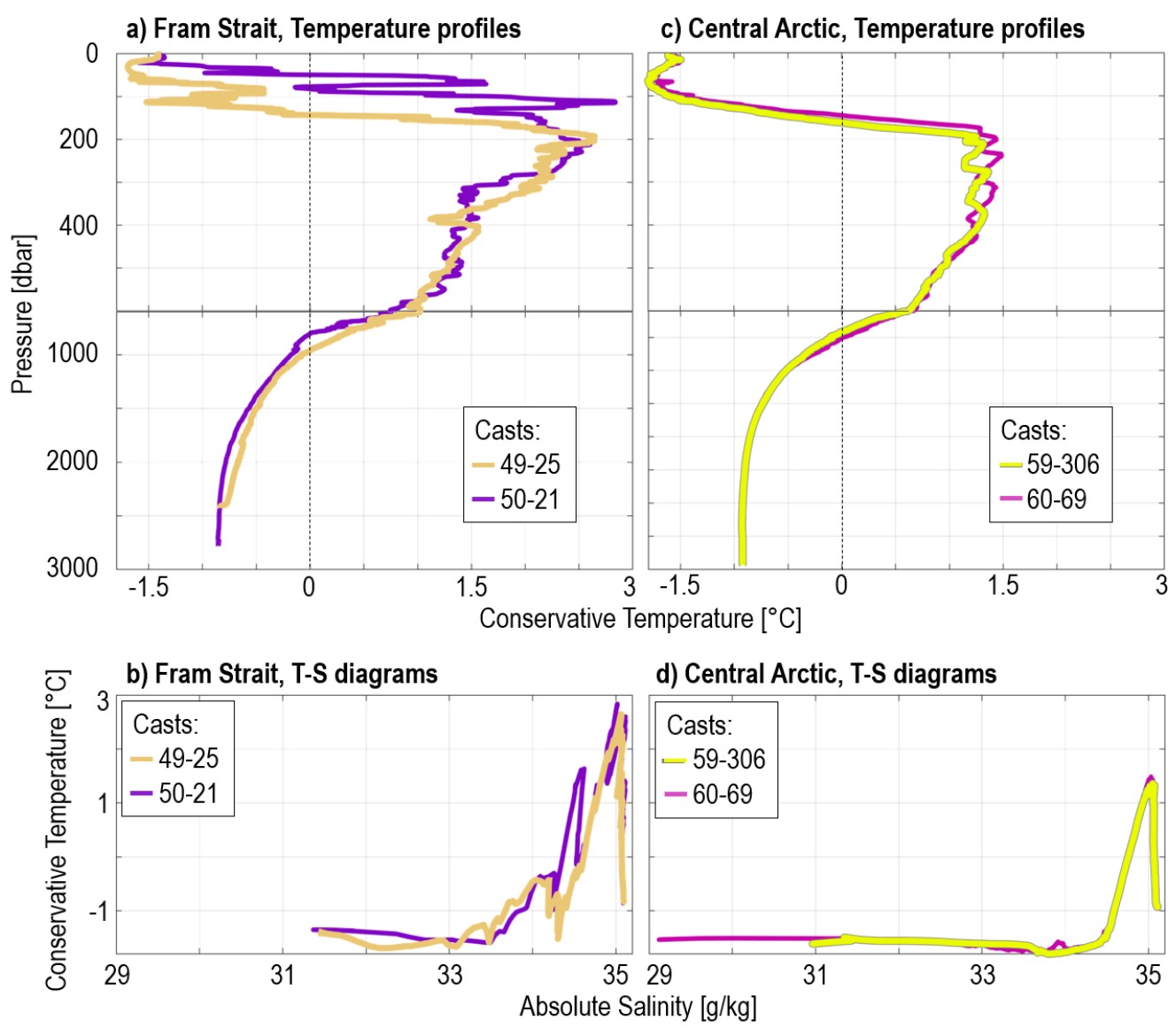

**Figure 6.** For the two Fram Strait profiles analysed in subsection 5.1 (left) and the two profiles of the Central Arctic analysed in subsection 5.2 (right), full-depth profiles of conservative temperature (top) and conservative temperature - absolute salinity diagram (bottom). Note the discontinued y-axis on the temperature profiles. Black vertical lines on a) and c) at 0 °C indicates the limits of the Atlantic Water. On the T-S diagrams, the ocean surface is to the bottom left and depth increases as the line moves towards increasing salinities.





## 5.1 Profile agreement in Fram Strait

The two profiles in Fram Strait have the cast numbers 49-25 (orange/yellow on all figures) and 50-21 (purple). They are 72 km
away from each other. 49-25 was collected on 29 July 2020 while the R/V Polarstern was still moored at the ice floe; 50-21
was collected on 5 August 2020 during transit. Tritium was collected only during cast 49-25. Helium, neon, and tritium were
collected in the upper 500 m only; CFC-12 and $SF_6$ to the sea floor, approximately 3000 m deep.

At the surface, the helium (neon) is lower (larger) for cast 49-25 than for 50-21 (Fig. 7a and b). The difference between
the profiles is even more striking when looking at the helium/neon ratio (Fig. 7c): The first cast 49-25 has a low value of
$\Delta(\mathrm{He/Ne})$ around -2% at the surface, steadily increasing for the rest of the profile, whereas the second cast 50-21 is positive
at the surface and more inconsistent throughout the profile. The negative values at the surface for 49-25 could be the result
of sea ice formation, although the air temperatures reported in Shupe et al. (2022) were hovering around 0°C in the week
leading to cast 49-25, which rather suggests sea ice melt. Verifying whether sea ice formation actually took place is beyond
the scope of this data paper. The increased helium value around 350 m in cast 50-21 (Fig. 6a, purple) could indicate the
presence of Greenland meltwater (Huhn et al., 2021), but is more likely not significant. Note that all values are within the range
observed during PS100 (grey dots), with somewhat higher values for neon during MOSAiC (Fig. 6b), further suggesting sea
ice formation.

[3]He is rather unremarkable for these particular casts and within the range observed during PS100 (Fig. 7d). There is no
consistent pattern for each profile, nor any consistent difference between the casts, which is not surprising given the many, very
strong intrusions observed in the upper 300 m in both profiles (Fig. 6). Similarly, there is nothing noteworthy in the tritium
profile (Fig. 7e), as expected in a depth range and location away from any source. It is worth noting that neon, [3]He and tritium
all have a sharp decline in the upper 200 m, i.e. as we transition into the Atlantic Water.

The CFC-12 and $SF_6$ values of cast 49-25 (yellow, Fig. 7f and g) are larger than those of cast 50-21 (purple) in the upper
200 m. As mentioned before, this first cast is also colder and fresher; the difference in concentrations can be explained by the
resulting higher solubility, and the difference becomes insignificant when comparing the partial pressures (not shown). Deeper
than 200 m and for the rest of the Atlantic Water (to approx. 800 m depth), no hydrographic profile is consistently warmer
or saltier than the other (Fig. 6a and b). Consequently, the CFC-12 values are almost identical for both casts in that depth
range, and the $SF_6$ values are similar, with no profile having consistently higher or lower values than the other. Finally, below
the Atlantic Water and down to the sea floor, the CFC-12 and $SF_6$ concentrations are lower for 49-25 (yellow) than for 50-21
(purple), and 49-25 is also warmer. However, the difference persists when comparing the partial pressure (not shown). The
differences are more likely caused by different water masses and/or different (re-)circulation, but more data would be required
to establish this. Although both variables, for both casts, are within the range of PS100 and GLODAPv2 data, it is worth noting
that $SF_6$ in GLODAPv2 is very noisy and that the data quality seems variable.



**Figure 7.** Two exemplary profiles collected a week and 72 km apart in Fram Strait in summer 2020 during MOSAiC for a) helium, b) neon, c) helium to neon ratio, d) ³He, e) tritium, f) CFC-12 concentration and g) SF₆ concentration. Grey dots are reference values from PS100 (pale grey) and GLODAPv2 (dark grey), if available. See locations on Fig. 4.



## 5.2 Profile agreement in the Central Arctic

The two profiles in the Central Arctic Ocean have the cast numbers 59-306 (bright yellow on all figures) and 60-69 (purple/magenta). They are 80 km apart. Both casts were collected while the R/V Polarstern was moored to the ice, 59-306 on 27 August 2020 and 60-69 on 3 September 2020. Tritium was collected only during cast 59-306. Helium, neon, and tritium were collected in the upper 500 m only; CFC-12 and $SF_6$ to the sea floor, approximately 4400 m deep.

The upper 100 m are very different for both profiles for helium, neon, and their ratio (Fig. 8a-c). Starting with cast 59-306

(yellow), we observe a zigzag pattern in helium, decreasing significantly from 5.8% at 10 m depth to 4.4% at 50 m, to increase again to 5% at 100 m. Changes in neon are less strong but follow the opposite pattern: increase then decrease. Consequently, He/Ne goes from nearly 0% at the surface to -1.4% at 50 m, and increases afterwards. The signal around 50 m could indicate a by-product of the previous autumn's sea ice formation, carried with the mixed layer in the previous winter when it was deeper, and now trapped below the shallow summer mixed layer (approx 10 m deep). A week later in contrast, cast 60-69 (purple) has

low He and Ne at the surface, which both increase below. The surface salinity is approx. 2 g/kg fresher for cast 60-69 than for cast 59-306 taken a week prior (Fig. 6d), which suggests that the ice has melted between the two casts and/or that the ship has drifted into different surface waters. Finally, the very different values between the two casts at 300 m (59-306) and 200 m (60-69) indicate that the samples happen to be taken in different waters, consistent with the large intrusions in that depth range (Fig. 6c).

The two casts do not show significant differences in the upper 50 m in the $^3$He signals (Fig. 8d), with differences within the measurement error range. For both casts, $^3$He then increases with depth, most likely as a result of tritium decay, as expected for that depth range in the Central Arctic Ocean.

At the surface, the water is at the same temperature (-1.5°C) in both casts but the salinity differed by 2 g/kg (Fig. 6d). Hence, we expect a solubility difference of the order of 0.1 pmol/kg for CFC-12 and 0.1 fmol/kg for $SF_6$, which is consistent with

the observed differences in concentration between the two casts at 10 and 20 m depth for CFC-12 (Fig. 8f) and at 10 m depth for $SF_6$ (Fig. 8g). The difference at 20 m in $SF_6$ is 0.35, or 3 times as high as expected from the solubility difference only; the corresponding density (not shown) indicates a small instability at that depth, so the gas deficit could be caused by mild overturning. At 50 m, the salinity difference between the two profiles decreases by a factor of 10 while the temperature remains similar; from solubility alone, the difference in CFC-12 between the two profiles should be 0.01 pmol/kg, but it remains at 0.1.

There is no $SF_6$ value at that depth, but in agreement with the strong stratification evidenced by e.g. the $\Delta(\text{He/Ne})$ signal, this difference might still reflect the solubility difference when the waters of these two profiles were at the surface. Below, in the Atlantic Water layer, the two profiles have somewhat constant and similar values in both CFC-12 and $SF_6$. Differences could come, as explained above, by sampling of different waters in the intrusions. Interestingly, in the bottom 1000 m, both profiles have values above the detection threshold in CFC-12 (Fig. 8f) but below the detection threshold in $SF_6$ (Fig. 8g). This

suggests that these waters were last at the surface in the 1930s, when CFC-12 were already used but industrial usage of $SF_6$ was still in its infancy. The historical CFC-12 data clearly show two different regimes, especially so in the upper 1000 m, with our profiles fitting nicely in-between (Fig. 8f). The historical $SF_6$ values are consistently lower than ours (Fig. 8g). Although



not specified on Fig. 4 for readability, not all GLODAPv2 profiles had both CFC-12 and $SF_6$, so that the reference $SF_6$ profiles are systematically further towards the central Amundsen Basin than ours. They also were collected 15 to 30 years before ours, and the reader should bear in mind that $SF_6$ is still increasing in the atmosphere. It is therefore no surprise that these reference profiles have lower $SF_6$ values than ours.

## 5.3 Brief comparison of the two regions

We now briefly compare profile 49-25 of Fram Strait with profile 59-306 of the Central Arctic, as they both have measurements for all tracers, including tritium. We chose summer profiles for both regions to try and minimise the effect of seasonality, but acknowledge that disentangling the temporal and spatial variability may not be straightforward for some applications. In particular, we do not comment further on helium and neon (Fig. 9a-c), as the differences between the two locations is most likely caused by seasonal sea ice processes, as we discussed previously. The profiles are also provided as a function of density in supp. Fig. A1.

The Central Arctic Ocean cast 59-306 has systematically larger values of tritium than the Fram Strait cast 49-25 (Fig. 9e), which is to be expected as the Central Arctic is closer to the sources of tritium: rivers flowing onto the Arctic shelf (e.g. Schlosser et al., 1994). Nothing happens aside from the expected decrease in value with depth, as expected from profiles that are not directly influenced by a river outflow.

The larger $^3$He values (Fig. 9d) for the Central Arctic Ocean than for Fram Strait are consistent with the larger tritium we just described. Besides, as the seafloor lies 4000 m away from our measurements, $^3$He is unlikely to come from the mantle. The larger $^3$He values suggest that the waters in the (upper) Atlantic Water are older in the Central Arctic than they are in Fram Strait. The lower values in $SF_6$ for the Central Arctic (Fig. 9g), which persist even when considering the partial pressure rather than the concentration, confirm that this water is older. This is consistent with the difference in hydrography (Fig. 6), as described at the beginning of this section: Although on the western side of Fram Strait, our Fram Strait profile contains young water. The tracers and hydrography therefore indicate a recirculation in Fram Strait (e.g. Hofmann et al., 2021), extending to the sampling location.



**Figure 8.** Two exemplary profiles collected a week and 80 km apart in the Central Arctic Ocean in summer 2020 during MOSAiC for a) helium, b) neon, c) helium to neon ratio, d) $^3$He, e) tritium, f) CFC-12 concentration and g) SF$_6$ concentration. Grey dots are reference values from GLODAPv2, if available. See locations on Fig. 4.







**Figure 9.** Comparison of the two profiles 49-25 (orange with black contour, in Fram Strait) and 59-306 (bright yellow with grey contour, in the Central Arctic Ocean), already shown on Figs. 7 and 8, respectively. This figure is also available with the density on the y-axis as supp. Fig. A1.





## 6 Summary

In this manuscript, we describe the CFC-12, $SF_6$, tritium, helium and neon data set produced from the samples collected between October 2019 and September 2020 during the MOSAiC expedition to the Eurasian Arctic Ocean. Noble gases and tritium were limited to the upper 500 m, whereas CFC-12 and $SF_6$ were collected for the full-depth. All tracers are available at
weekly or higher temporal resolution, although CFC-12 and $SF_6$ are limited to the upper 1000 m during the two months period (March - May 2020) where the ship CTD could not be operated. We showed that individual tracers can be used or combined with each other to investigate rapid sea ice processes, (suspected) ocean mixing, and even the presence of oceanic recirculation branches. By studying them in relation to sparse, previously collected data, or other tracers, they can be used to study the large scale oceanic circulation or even elucidate the impact of climate change on ventilation.

Unsurprisingly, the main conclusion for us is that we regret not collecting more samples. Having a larger team and/or more experienced samplers would have allowed us to collect tracers at a higher vertical resolution. The ice dynamics made the ship CTD inoperable during two months, which coincided with the transition from the Amundsen to the Nansen basin via Gakkel Ridge. This possibility had been foreseen during the MOSAiC planning phase, so an alternative water collection system via the moonpool had been devised, but ultimately not implemented as it was too expensive. Besides, due to the objectives of the
contributing projects, the funding for noble gases and tritium, which can be used to track hydrothermal plumes and ventilation, respectively, was limited to the upper 500 m.

As the Arctic, a hotspot of climate change, becomes the focal point of many teams and funding agencies, we strongly recommend that future endeavours collect samples of these tracers all the way to the sea floor, especially so in the vicinity of Gakkel Ridge and close to suspected overflow locations (listed in e.g. Luneva et al., 2020).

## 340   7   Data availability

The complete MOSAiC tracer data set has been submitted to PANGAEA as Huhn et al. (2023); DOI is under registration and will be added during the review process. In the meantime, the data set is available as a single .zip as supplemental material. The MOSAiC CTD data sets are freely available on PANGAEA as Tippenhauer et al. (2023a, data set doi: 10.1594/PANGAEA.959964, last accessed 21 June 2023) and Tippenhauer et al. (2023b, data set doi: 10.1594/PANGAEA.959963, last
accessed 21 June 2023). The IBCAO bathymetry of Jakobsson et al. (2020) is freely available via www.gebco.net/. The PS100 data used in section 5.1 are freely available on PANGAEA via https://doi.org/10.1594/PANGAEA.931336 (data set DOI:10.1594/PANGAEA.931336, last accessed 29 May 2023). The GLODAP2022v2 data set is freely available via https://www.ncei.noaa.gov/data/oceans/ncei/ocads/data/0257247/ (data set DOI:10.25921/1f4w-0t92, last accessed 23 May 2023).



**Figure A1.** Same as Fig. 9 but as a function of the potential density referenced to 2000 dbar $\sigma_2$.



*Author contributions.* Initial conception of the study: CH, OH and MW. Fielwork logistics and/or collection of samples: JA, Y-CF, HH, CH,
SK, DK, IK, MM, CM, BR, NS, ST. Lab analysis of the samples: KB, OH, JS. Preparation of the original draft, including visualisations: KB,
CH, OH, SK, WK, JS, MV, MW. Initial submission: all

*Competing interests.* The authors declare no conflict of interest.

*Acknowledgements.* This work was carried out and data used in this manuscript were produced as part of the international Multidisci-
plinary drifting Observatory for the Study of the Arctic Climate (MOSAiC) with the tag MOSAiC20192020 (AWI_PS122_00). We thank
all those who contributed to MOSAiC and made this endeavor possible (Nixdorf et al., 2021). CH and SK are funded by Vetenskapsrådet
grant number 2018-03859 awarded to CH, project "Why is the deep Arctic Ocean Warming? (WAOW)", and acknowledge support from
the Swedish Polar Research Secretariat for berth fees onboard MOSAiC. WK, NS, and MW gratefully acknowledge the funding by the
Deutsche Forschungsgemeinschaft (DFG, German Research Foundation) – Project Number 268020496–TRR 172, within the Transregional
Collaborative Research Center "ArctiC Amplification: Climate Relevant Atmospheric and SurfaCe Processes, and Feedback Mechanisms
$(AC)^3$". This work contributes to the Changing Arctic Ocean (CAO) program, jointly funded by the UKRI Natural Environment Research
Council (NERC) and the BMBF, project Advective Pathways of nutrients and key Ecological substances in the ARctic (APEAR) grants
NE/R012865/1, NE/R012865/2 and #03V01461; and the BMBF project Eddy Properties and Impacts in the Changing Arctic (EPICA),
#03F0889A. HH is funded by Chinese Polar Environmental Comprehensive Investigation and Assessment Programs.



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
