# Peer review of "A year of transient tracers chlorofluorocarbon 12 and sulfur hexafluoride, noble gases helium and neon, and tritium in the Arctic Ocean from the MOSAiC expedition (2019-2020)"

_Earth System Science Data, 2023_

## Author Response (AR1)

We thank the reviewers for their comments. Note that the contribution of the reviewers has been added to the acknowledgement section. We have addressed all their comments, as detailed below, starting with R1, then R2 from page 3 onwards.

**R1:**

**Line 13 states "the full-depth AO is changing", while at line 19 the statement is "we do not even know whether there is a change (below 1000m)". These two statements are contractionary. Please rephrase.**

Changed the first sentence to "The Arctic Ocean is changing rapidly"

**Lines 75-85. It appears that the AW is not of interest!**

We rephrased to emphasise that the AW was included in both aims and was, in fact, the water mass we were most interested in.

**Lines 121-126. This section is very descriptive and maybe too detailed on technical issues, suggest reformulating.**

Following this reviewer's and R2's comments, we deleted this subsection and moved its most relevant points either to the sampling subsection (now 2.2) or to the conclusions.

**Lines 175-179: The work by Bulsiewicz (1998) reports on up 10% of CFC12 contained in the headspace, but do not report on SF6. Please add this information as this might be important for understanding of the reproducibility of the SF6 measurements.**

The measured loss into the headspace has been added to section 3.3 for both CFC-12 and SF6.

**Table 1: Why not have salinity and temperature in the data set directly? The tracer data are pretty much useless without S/T (P is included). Including S/T would save the user the trouble to find the CTD data and combine the files.**

The CTD data belong to different data sets (one for the ship CTD and one for the Ocean City CTD), produced by someone else. Republishing this person's data would not be good scientific practice, especially so when they have made it clear that they do not consent. Both our data product and manuscript cite the CTD datasets and provide links to them, so we trust that the interested user will access them in two clicks.

**Data: It would be easier to work with the data if a number was used for the snow data, rather than using letters in the data file.**

PANGAEA in fact required that we submit the snow and the ocean data as two different products. The text has been modified accordingly.

**Line 190: Will need a direct link to the data set, and one to the MOSAIC data set in general. I was not able to find the tracer data at Pangaea.**

The data sets were under review. Links to the MOSAiC CTD datasets were already available but more iterations have been added throughout the text.

**Figure 4: Do some of the gray dots in the map refer to tracer samples discussed in Stöven et al 2016 (doi:10.5194/os-12-319-2016). If so, please refer to this work.**

Following a later comment by the reviewer, we have modified, moved, and shortened this section. We have added a reference to the publication recommended by the reviewer.

**Polarstern cruises are often known by acronyms like "ARK-XXVII/1". Can you provide this reference for "ps100" as well to facilitate the finding of data.**

Now provided (ARK-XXX/2)

**Section 5.1: The Fram Strait is well-known for variability in both space and time. I wonder what the significance of this whole section is in terms of understanding the data presented in this ms?**

Following this reviewer's and R2's comments, we have modified the manuscript so that most of the profile comparison focusses on the more-slowly-varying Central Arctic. The Fram Strait subsection has been drastically shortened, and moved after the Central Arctic one. Note that the objective of this subsection is not to understand the data – this is a data description paper, not an original scientific study. The features that we discuss only highlight possible usages of the data set.

**Lines 333-334: Is it relevant to refer to something that was planned, but never happened? In particular in the conclusion section.**

Yes, we argue that is an important lesson learnt and a warning to other programs and future expeditions. See also comment from R2 in the same direction as ours.

**Section 5: Please refer to the individual data sets, if possible, rather than generically to "GLODAP".**

We disagree with the reviewer. GLODAP(v2) is more than a collection of individual data sets, it also includes homogenisation and quality control. Therefore here, we are not using the original datasets, but their GLODAP-absorbed version.

**R2:**

**General comments:**

**This paper offers a clear example of the strength of chemical tracers (whether from natural or artificial origin) and how they can be utilized to comprehend specific oceanic processes. This applies to both local and global scales, as well as shorter and more extended timescales. I would suggest the authors to motivate this a bit better in the introduction, and give a more comprehensive explanation of the added value of combining these 5 gas tracers. For example: the combination of He and Ne can tell about sea-ice processes, but what are the range of values that one would expect and which values would tell about more/less sea-ice formation? Is Helium and Tritium going to add something to this story, or will look into other processes? Which ones exactly and also at the surface, on only at depth (where hydrothermal activity is expected)? And finally, what is motivation of looking into CFC-12 and SF6 and how does it add/complement the results obtained from the other three gases? Why He, Ne and Tritium will be only looked at surface/shallow waters (down to 500m) and CFC-12 and SF6 down to 4000m?**

We added specific values when possible. In response to this and the next comment, we added a paragraph discussing the possible applications of combining several tracers, including references to people that have combines them, usually to derive an age of water, but also by making use of their different solubilities and equilibration times. We also explain why, scientifically, one would look at He, Ne, and Tritium only in the upper water column. The real reason, which we do not indicate in the manuscript, is however more basic: money. We could not afford to sample the entire water column, no matter how much we would have wanted to.

**Maybe there could also be some information in the introduction about the gas saturation that is expected for those tracers and how these can change depending on different environmental conditions (e.g. temperature). Are some of these gases more sensitive to changes in temperature and therefore could trace certain processes better than others? Or, are the equilibration times between atmosphere and sea surface similar to all of them? Which ones are faster and thus would be targeted to trace faster changes? Not being an expert on this, such information would be very useful…**

In response to this and the previous comment, we added a short text in the introduction. The derivation of the values is provided in a new appendix text A1.

**Figure 2 is very schematic but also very illustrative, as it brings together the processes and the tracers. Most probably, this figure will see extensive utilization for educational purposes. However, I would highly recommend to merge the text above the figure within the figure itself, so adding the tracer information next to each process (maybe using different colors or font type) and even a range of expected values that would reflect on the extent of each process.**

The figure has been modified so that there no longer is any text above the figure; everything has been merged as suggested by the reviewer. There is unfortunately no such thing as a range of expected values. For each process and each tracer, even the direction of the change will depend on a range of parameters. For example for the first process "glacial meltwater",

the values will depend on the location on the glacier, in particular the distance from the calving front.

**Figure 3 shows an x-axis that is time. Why not stations/basins? Is the purpose of this to look into seasonal variability from results? I find Figure 4 more illustrative. Could these two figures be merged into a single figure to explain both the geographical locations of samples and the corresponding sampling times.**

Yes, the purpose of this figure was to indicate the temporal distribution, for readers that are interested in studying for example the seasonal variability, whereas that of figure 4 is to indicate the geographical distribution, for readers interested in the spatial variability. We have added a sentence to clarify this in the text. However, note that we, in this manuscript, do not study anything specific: This is a data description paper, not original science. We simply suggest what others could do with the data.

**I found section 2.2 "challenges encountered in the field" interesting to read, but I am not sure that it is so necessary to have it in the main body of the paper. Did these challenges really hinder some of the results you wanted to have? Or it only wants to illustrate the difficulties encountered during the expedition and justify the lack of some data that would have helped with the interpretation of results? My suggestion would be to bring such a message at the very end of the paper as a motivation to get more data in the future and cover the existing gaps.**

The objective of this subsection was both to explain gaps in the data collection and to act as a "lessons learnt" reflection for future expeditions. In response to this comment and a similar comment from R1, we removed this section. The most relevant sentences have been moved either to the sampling subsection or to the conclusion.

**Section 5.1 compares profiles taken in two different locations (Fram Strait and Central Arctic) and in two different casts at each location. If I understood that correct, the reason to look at two different casts is to actually compare changes in tracer distribution at very short timescales. However, this is not really stated like this and thus a bit confusing. Could the cast numbers be changed for "day 1" and "day 6" to represent the time span between 29th of July 2020 and 5th August 2020, for the Fram Strait? Also, these two casts are taken at locations that are 72 km away from each other. How much the changes observed in He and Ne at surface could be caused by a slightly different location? Or, in other words, how representative are these two casts of a location that is affected by the same processes despite being 72 km a part? (a similar comment would apply to section 5.2).**

In response to this comment, the next two comments of the reviewer, and the more blunt comment of R1 on the "limited value" of this subsection, we have modified the manuscript. Note that the objective of section 5 is to show possible usages of the data set, not to conduct a proper scientific study. This section now starts with the profile comparison in the Central Arctic Ocean, and only briefly compares profiles in Fram Strait in subsection 5.2, which is a shortened version of the earlier Fram Strait subsection. Throughout the now-main subsection on the Central Arctic Ocean, we have added text to remind the reader that the two profiles discussed have been collected one week and 80 km apart, when relevant.

**Line 239: "… the profiles are even more striking when looking at helium/neon." This is probably a comparison to the changes observed only when looking at helium and neon from the two casts. Yet, the text is a bit vague. I would first expect to read some explanation why the helium and neon are different between the two casts and why this was not-/expected?**

See previous comment: This sentence has been removed.

**Lines 248 – 253: What is the added value of looking into 3He and tritium? As it is written here it looks like there is no consistent pattern and also that no pattern was actually expected…? This links to one of the previous comments. If authors could better describe what is expected from these two tracers, then it would also be easier to interpret what is shown in figures 7 and 8.**

See two comments prior: This sentence has been removed. See also response to the comment recommending that we add such explanation to the introduction (which we did).

**Regarding the interpretation of 3He, there is no reference to whether this isotope is coming from the decay of tritium or whether it is of natural origin. How can these two sources be distinguished from one another?**

We have added to the introduction a sentence and reference to address this comment. Short answer is: these two cannot be distinguished without additional information.

**The two anthropogenic gases CFC-12 and SF6 have been long used to understand ventilation processes. In the Arctic Ocean there are a few studies that use the combination of this tracer pair to describe transport timescales and ventilation rates of Atlantic waters (e.g. Smith et al., 2011; Smith et al., 2022). Other studies use the tracer as a tool to estimate the anthropogenic carbon storage in the Arctic Ocean using the so-called transit time distribution model (e.g. Tanhua et al., 2009; Rajasakaren et al., 2019; Terhaar et al. 2020). In this study, there is no reference to the potential of using these two tracers for such purpose. I understand doing this exercise could be beyond the scope of this paper, but I would encourage the authors to acknowledge these points for which these new data can be used in future studies.**

This point was already present in the introduction. Following one of the first comments made by the reviewer, we added more text about TTD and age of water to the introduction.

**Regarding the 3H sources, authors mention "rivers flowing onto the Arctic shelf (Schlosser et al., 1994). But, what about other sources of Tritium? What are the amounts of 3H released from La Hague? Could this signal also enter the Arctic Ocean as it is the case for other radionuclides of nuclear origin (e.g. 137Cs), and what would be its presence in the central Arctic and Fram Strait if this source was significant compared to rivers?**

We have added a sentence about this in the introduction. Short answer is: no, this is insignificant. The first reason is the physical oceanography: surface waters from the Channel do not reach the Arctic. Depending on the coastline, they are either advected southward into the Bay of Biscay and continue travelling south, or they are advected north into the North Sea. Once in the North Sea, they need to undergo a lot of modifications to join the deep

waters that will flow into the Arctic. That is, they have to be strongly diluted, at which point their Tritium content will be negligible.

Neglected the physical oceanography aspect, one can also do a back of the envelope calculation. Both Wikipedia and the Bailly du Bois data set on PANGAEA (https://doi.pangaea.de/10.1594/PANGAEA.906749) yield approx. 10^16 Bq/a for La Hague. Divided by the volume of the Arctic mixed layer (ca 10^7 km2 x 100 m), this results in 10^-2 Bq/L, or < 0.1 TU. That is, if the water from La Hague entered the Arctic without dilution, its Tritium would be undetectable at the pan-Arctic scale.

There is one possible local source of Tritium release that we cannot account for: nuclear powered submarine accidents. We added a reference that says that "under normal operations", the tritium released by their reactor is unsignificant though.

**When comparing the profiles of 3He of the Central Arctic and Fram Strait, authors see that 3He values are larger in the central Arctic than in the Fram Strait. One of the hypotheses is that surface waters in the Central Arctic are older than in the Fram Strait. But, what does this age refer to? Is this a transport time from a certain location to another? or a ventilation rate? What's the age exactly referring to? Or, maybe it refers to the input of 3He into the marine system, so that what they observe in the Fram Strait is a fresher input of 3He compared to the one they see in the Central Arctic? I believe this concept could benefit from further clarification.**

This sentence refers to the last paragraph of section 5.3. We have rephrased to clarify that this is not about the surface waters, but the waters deeper than 2-300 m, i.e. the Atlantic Water. We have changed the order of the sentences to more clearly link the age to both 3He and SF6. Finally, we have also rephrased to clarify that by "older", we mean the ventilation / transport from the source (at the surface of the North Atlantic).

**CFC-12 and SF6 have been introduced to the surface of the ocean following their industrial releases (one should also cite RA Fine, 2011). The temporal changes of CFC-12 and SF6 in the water column when comparing to historical data are mostly related to this transient nature of this gas tracers. Figures 7 and 8 (panels f and g) would benefit from having time information (year when samples were collected). I believe PS100 happened in 2015, but what about the data taken from GLODAPv2? It looks like some should be previous to 2015 and other that was taken after 2015…**

The reference Fine (2011) has been added to the introduction. We have modified the figures; the years are now indicated. As can be seen on the revised figures, all GLODAPv2 profiles shown here were collected latest in 2015.

**Specific comments:**

**Figure 1 could show other important features that are also described in the introduction. For example, the Beaufort Gyre, the Canada Basin, Lomonsov Ridge. I think this would give a more complete figure of the Arctic Ocean, even though the study area is only focused in the Eurasian Basin**

The figure has been modified as suggested by the reviewer.

**Line 30: other references on previous studies using gas tracers in the Arctic Ocean are Smith et al., 2022; Tanhua et al., 2009; Smith et al., 2011; Rajasakaren et al., 2019.**

The references in the manuscript referred to the sentence "beginning of modern-day research". We have modified the sentence to indicate that it is still done nowadays. Note that all but one of the references suggested by the reviewer were already cited in the text. The missing one has been added.

**Line 52: change precipication to precipitation.**

Done

**Line 82. For the sampling strategy of the deep ocean, SF6 and CFC are not ideal tracers. Authors mention that futures efforts should also sample tritium, but then I see this could only be a tracer of "deep ocean" wherever there is a hydrothermal plume (so basically only on Gakkel ridge). Why not thinking of other tracers of deep waters such as 14C (or even 39Ar)?**

We have added a sentence to address this comment to the introduction. The honest answer though is that we are no expert in these other tracers. We do collaborate with people that were doing carbon chemistry, but those colleagues require CFC and SF6 to constrain their carbon values. As for 39Ar, there was a very big "water pressure" on the CTD during MOSAiC due to the large ecosystem team. Although the latest method needs only 5 L, that is still 20 times more water than the CFC samples and would have been refused. In the future, we do hope that we can routinely obtain 39Ar.

**Maybe Table 1 contains information that maybe it is not so necessary for the purpose of the paper and could be moved to supplementary material?**

This comment is too vague for us to understand which information the reviewer finds unnecessary. Obviously, we included only information that we, as data users, would want to see. We have not changed anything, aside from what was induced by PANGAEA separating our joint data set into one for the ocean and one for the snow.

**Line 165: Change "than" to "then"**

Done

**Line 250: to which intrusions does this refer to?**

See earlier comments: This sentence has been removed.

---

## Referee Report (RR1)

The paper "A year of transient tracers chlorofluorocarbons 12 and sulfur hexafluoride, noble gases helium and neon, and tritium in the Arctic Ocean from the MOSAiC expedition (2019-2020)" has been carefully revised by the authors and resulted in a great improvement of the manuscript. I would like to thank the authors for incorporating my previous comments into the new version, and acknowledging it. I would accept the manuscript as is, but here I wanted to give some last minor comments/thoughts, in case the authors want to consider them for the final version of the manuscript.

Technical/specific comments:

Line 70: Change transient time distribution to Transit Time Distribution (TTD).
Same Line 70: I would rephrase this sentence to "… by first computing the transit time distribution (TTD), which is a measure of the age spectrum of a water mass (Tanhua et al., 2009)". Here it would maybe consider to refer to the papers from Waugh, Hall, etc.

Table 1 and Table 2 are very similar, but Table 2 has much less parameters than Table 1. Just in case the authors want to merge this two, they could mark the 10 parameters in Table 2 (e.g. with a *) in Table 1. Alternatively, one could add an extra column in Table 1 indicating which datasets (i.e. ocean/snow) include each parameter.

Figure 7 has greatly improved with the addition of sampling years. However, for the CFC-12, I can't see any datapoint collected in 2015. Is there no CFC-12 data that was collected in 2015? There should be from PS94... but maybe there is a reason not to include them?

Figure 8f would benefit from also adding the year of PS100 in the legend, as 2016 would cover the gap between 2012 and the new data of MOSAiC (2019 - 2020). Is there no SF6 data from PS100? This would also be very valuable to have, as one could use the combination of CFC-12 and SF6 published in this work, to estimate the TTDs using the gas tracer-pair.